# A Follow-Up Study of Cognitive Development in Low Risk Preterm Children

**DOI:** 10.3390/ijerph17072380

**Published:** 2020-03-31

**Authors:** Miguel Pérez-Pereira, María Pilar Fernández, María Luisa Gómez-Taibo, Zeltia Martínez-López, Constantino Arce

**Affiliations:** 1Department of Developmental and Educational Psychology, University of Santiago de Compostela, 15782 Santiago de Compostela, Spain; zeltia.martinez@usc.es; 2Department of Psychology, University of A Coruña, 15190 A Coruña, Spain; pilar.fernandez1@udc.es (M.P.F.); marisa@udc.es (M.L.G.-T.); 3Deparment of Social, Basic and Methodological Psychology, University of Santiago de Compostela, 15782 Santiago de Compostela, Spain; constantino.arce@usc.es

**Keywords:** preterm children, cognitive development, predictive factors, longitudinal study

## Abstract

The results of a longitudinal study on the cognitive development of one group of full-term and three groups of low risk preterm children with different gestational ages (GA) are presented. The 181 participants were divided into four GA groups of similar size. The aims were: 1) To check if there are differences in cognitive development (measured through the Batelle scale) among the GA groups. 2) To establish the predictive factors of cognitive development at 22 and 60 months of age, taking into account biomedical, environmental and individual factors. The results of the repeated measures ANOVA performed at 22 and 60 months of age indicated that the cognitive trajectories of the four GA groups were similar. Linear regression analyses showed that the effect of the different predictors changed in relation to the time of measurement of cognitive development. Biological factors and the quality of home environment had a moderate effect on the cognitive development at 22 months of age. Cognitive results obtained at 22 months of age, and, to a lesser extent, working memory had the greatest effect on cognitive development at 60 months. GA does not predict cognitive development. Preterm children do not show cognitive delay if they are healthy.

## 1. Introduction

Cognitive measures of children should be taken early to detect any developmental delay or differing patterns of development so that intervention programs may be provided. For this reason, follow up studies of the general cognitive abilities of preterm (PT) children in conjunction with those bio-medical, environmental and personal factors associated with cognitive outcomes are of great relevance [1]. In this introduction we will first review previous studies on cognitive development of preterm children. Secondly, we will identify those factors which predict cognitive development from the literature review.

Evidence exists that preterm children may show cognitive impairments in different domains when compared to full-term children [2]. A wide spectrum of cognitive impairments are observed among very preterm or very low birth weight (BW) children [3].

Gestational age (GA) is associated to cognitive development in such a way that the lower the GA, the worse the cognitive performance of preschool and school age PT children [4,5,6]. Differences between full-term and preterm children are particularly high between extremely preterm (EPT) or very preterm (VPT) and full-term (FT) children [3,7,8,9,10,11,12,13,14,15,16,17,18,19,20,21,22,23]. There is no unanimity, however, among the studies carried out with EPT or VPT children, and heterogeneity of effect size across studies is important [21]. A few studies did not find such dramatic differences between these children and FT children [24,25,26,27,28,29].

Moderate (MPT) or late (LPT) preterm birth, however, seems to affect cognitive development in a lesser way. Yaari et al. [30] observed that the trajectories of cognitive development of MPT were more favorable than those of EPT and VPT children, although of higher vulnerability in relation to FT children. Cognitive performance of MPT or LPT children between 2 and 7 years of age of age was found to be lower than that of FT children [31,32,33,34,35].

In contrast with the above reported results, a review study [36] and several investigations [37,38] did not observe significant differences between LPT and FT children of different ages. Other studies carried out with healthy LPT children did not find differences with FT children or with normative normal range at preschool ages [39] or between 4 and 15 years of age [40]. Even low risk preterm children with GA below 32 weeks (VPT) without major health problems obtained cognitive scores in the normal range, although significantly lower than those of the group of full-term children [41].

These latter results seem to indicate that healthy condition is an important factor which prevents cognitive delay. The need of new studies is particularly relevant given the dearth of longitudinal studies on cognitive development of low risk preterm children [31].

Different factors were suggested as predictors of cognitive outcomes in PT children. We will first comment on biomedical factors. GA (and BW) were considered the most relevant predictors of cognitive development [5,42,43], although other authors did not find this relationship to exist [27]. Prematurity probably interacts in a dynamic way with genetic, bio-medical, and environmental factors throughout development [8,14].

Major brain pathology was found to affect low cognitive level [13,17], and neonatal white matter abnormalities (in addition to family socioeconomic adversity) predicted cognitive risk in VPT children [16,44]. Bronchopulmonary dysplasia was found to be a crucial factor for cognitive outcome [21].

Further research is needed to determine if length of stay in neonatal intensive care unit (NICU) is an independent predictor of poor neurodevelopment outcome [45,46]. Evidence of the influence of the Apgar scores on cognitive outcomes is confusing. Apgar tests are administered to newborn children in the first minute of life (and also at minutes 5 and 10, if necessary) and provides a measure of the health state of the children in five vital areas, offering a score from 0 to 10. Children with low Apgar scores were more likely to have low IQ score at 18 years of age [47], however, unit increase in the Apgar score at 5 minutes was not associated with significant decreases in vulnerability on the language and communication domain at 5 years of age [48].

Male sex and non-white ethnicity (together with low parental education and low BW) were predictive of cognitive impairments in VPT at 5 years of age [13,49].

Maternal prenatal smoking has detrimental effects on prenatal brain development [50], is a risk for suboptimal cognitive and neuropsychological development in preterm infants [51,52], and was consistently associated with decrements in test scores of academic performance [53], and increased risk of poor scholastic achievement [54]. Maternal smoking habits has also been found to have a negative effect on cognitive development at 4 years of age [55], although this effect has been called into question [56]. Mother’s age at birth was associated with increasing cognitive vulnerability in a population-based cohort study [57].

As children grow older, however, environmental factors and social context seem to explain the greatest part of variation in cognitive development, and the influence of perinatal risk factors appears to diminish over time [49,58,59,60,61].

Socioeconomic status (SES) seems to be a prevalent factor related to cognitive development of EPT children. The risk of cognitive deficits seems to rise under adverse socioeconomic conditions [16,18,62], and the opposite seems to be true too: upper income level acts as a protection factor for EPT children [5]. Wong and Edwards [63] discovered that the mother’s educational level was the most highly valued indicator of socio-economic status as well as the factor that was most consistently associated with cognitive development in PT children. Low parental education was found to predict global cognitive impairment in VPT children [5,49]. Other studies corroborated that parents’ educational level is the best predictor of intelligence in PT children with different ages and prematurity levels [13,41,64,65,66,67].

Other environmental factors such as parental styles [58,59,68], maternal stimulation [69], employment quality and educational level of parents [70], marital status [71], do have an effect on the cognitive development of PT children.

Different investigations, carried out using the HOME scale to measure the quality and level of stimulation of the home environment, indicate the influence of the home environment on the cognitive development of PT children [27,72]. These results are of particular interest for our study, since we used this instrument.

In other investigations, the effect of individual or psychological variables was studied. The cognitive development achieved at earlier ages (associated or not to other factors) was found to be a predictor of cognitive outcomes [39,73,74,75,76]. These results point to a stability in cognitive measures throughout time, supporting the results found by Mangin et al. [16].

PT children, particularly VPT and EPT children, were found to perform worse than FT children in working memory tasks [70,77], which in turn affects cognitive achievements.

There is a dearth of longitudinal studies on the cognitive development of low risk PT children whose cognitive achievements and developmental trajectories may differ from other high risk and/or EPR or VPT children who have a higher probability of having biomedical problems.

The present investigation has the following aims:(1)To check if there are differences in cognitive development among groups of healthy children of different gestational ages.(2)To establish the predictive factors of cognitive development at 22 and 60 months of age, taking into account biomedical, environmental and individual factors.

The hypotheses of the study are:(1)Given the low risk nature of the sample we do not expect to find differences in cognitive development between the GA groups.(2)Biomedical factors will have a higher effect on early cognitive development, and their influence on cognitive development will vanish later.(3)Environmental factors will increase their effect on cognitive development as children grow older.(4)There will be an influence of previous cognitive achievements on later cognitive development.

## 2. Materials and Methods 

This study was part of a longer longitudinal project.

### 2.1. Participants

There were 181 children of different gestational ages who were assessed at 22 months of age and were followed up to the age of 60 months. The children were classified into 4 GA groups: 1) very preterm (VPT) and extremely preterm (EPT) children with GA of 31 weeks or below; 2) moderately preterm children (MPT) with GA between 32 and 33 weeks; 3) late preterm children (LPT) with GA between 34 and 36 weeks; and 4) full term children (FT) with GA of 37 weeks or above. We intended that the number of participants in each group was similar for analysis purposes, given the scarcity of children EPT. Obviously, there was a loss in the number of participants from 22 to 60 months (see below).

The group of PT children did not show any additional serious complications. Children were excluded from the sample if they had any of the following characteristics: cerebral palsy, periventricular leukomalacia, intraventricular hemorrhage higher than II, encephalopathy, hydrocephalus, genetic malformations, chromosomal syndromes, metabolic syndromes associated to mental retardation, severe motor or sensorial impairments, or Apgar scores below 6 at 5 min.

The characteristics of the PT children together with the exclusion criteria used allow us to consider the sample as a low risk PT group.

At 22 months of age, the 138 PT children had a mean GA (and SD) of 32.62 (2.41), a mean BW of 1721.70 (435.36), and a mean Apgar score (first minute) of 7.94 (1.30). As for the 43 FT children, they had a mean GA of 39.70 (1.48), a mean BW of 3373.83 (433.09), and a mean Apgar score (1st minute) of 8.13 (1.20). Maternal education has been categorized into three levels: 1) Basic maternal education (primary and secondary education), 2) High school and technical school and 3) University degree.

Stay in NICU was categorized into three levels 1) No stay, 2) 1-15 days, 3) over 15 days.

The characteristics of the four gestational age groups and chi square and one factor ANOVA comparisons among them are shown in Table 1.

There were no differences between the four GA groups in most of the variables displayed in Table 1, except for stay in NICU, mother’s age, and Apgar score in the first minute. As for the results of the chi square test, the PT children presented longer stays than FT children, as expected. The higher the degree of prematurity, the longer the stay in NICU. In the case of the ANOVA, Bonferroni post hoc analysis indicated that significant differences (*p* < 0.05) in Apgar score between the GA group of ≤31 weeks and the other 3 groups (32–33, 34–36 and ≥37 weeks of gestational age) were responsible for the results. The group of children with GA ≤31 weeks obtained significantly lower Apgar scores than the other groups.

In the case of mother’s age, significant differences were found between the group of children with GA ≥37 weeks and the groups of children with gestational ages of ≤31 and 32–33 weeks. The group with GA ≥37 had mothers significantly younger than the others.

Although the number of participants varied at each point of measurement, the characteristics of the PT sample did not vary throughout the period studied. The number of participants at each point of cognitive ability measurement appears below.

### 2.2. Procedure

Parents’ consent, and approval by the Galician Ethics Committee of Clinical Research (code 2008/010) were obtained before the beginning of the investigation. The children were assessed when they were 22, 48 and 60 months of age. Corrected age for PT children was used at 22 months of age. The following instruments were administered at the indicated ages.

### 2.3. Instruments

When the children were 22 months of age and 60 months of age, the Spanish version of the Batelle Developmental Inventory (BDI) [78] was administered to assess cognitive development. The skills assessed by the BDI scale are adaptive, personal-social, communication, motor, and cognitive. The cognitive raw score was used for the analysis. The cognitive score is composed of the following elements: perceptive discrimination, memory, reasoning and school skills, and conceptual development.

The Home Observation for Measurement of the Environment (HOME) [79] in its Spanish version [80] was used. The HOME is designed to measure the quality and quantity of stimulation and support available to a child in the home environment. The information is gathered through personal interview to mothers and through observation of the children’s homes and the interactions between the children and the mothers during the interview session. Higher total HOME scores indicate a more enriched home environment. The version for infants and toddlers (0–2 years) was administered when the children were 22 months of age. This inventory is composed of 45 items that are presented as statements to be scored as YES or NO. The HOME includes six subscales: Emotional and verbal responsiveness of the primary caregiver, Avoidance and restriction of punishment, Organization of the physical and temporal environment, Provision of appropriate play materials, Parental involvement with the child, and Opportunities for variety in daily stimulation. When the children were 48 months of age, the HOME scale for preschool children (3–5 years) was administered. This inventory is composed of 55 items that are presented as statements to be scored as YES or NO. The HOME includes seven subscales: Learning materials, Linguistic stimulation, Physical environment, Tenderness and affection, Academic stimulation, Modeling, Diversity of experiences and Acceptance.

The total score of the two HOME versions (for infants and toddlers and for preschoolers) was used for the analyses. The results obtained by the four GA groups in the two HOME tasks are offered in Table 1.

The CORSI ordering task [81,82] was used to assess the non-verbal working memory when the children were 60 months-old. Colored blocks are highlighted in a given sequence. The children must repeat the sequence. The total raw score was used for the analysis. The results of the Corsi task are shown in Table 1.

In addition, a sociodemographic and health interview was applied to the mothers shortly after the child’s birth, and biomedical information was collected from medical records. Information on gender, gestational age, length of stay in NICU, Apgar score at 1st minute, mother’s age, number of cigarettes smoked during pregnancy per day and mothers’ educational level, among other factors, was gathered.

### 2.4. Analysis Performed

First of all the data gathered with the BDI at 22 and 60 months of age were analyzed using a 2 (age) × 4 (GA groups) repeated measures ANOVA in order to test if there were intra subjects differences (age related differences in the same participants), inter subjects differences among GA groups and a combined effect age × GA group.

A linear regression analysis was performed using BDI cognitive score at 22 months of age as dependent variable (DV). As independent variables (IV), gender, gestational age (numerical), stay in NICU, Apgar score at 1st minute, mother’s age, and number of cigarettes smoked during pregnancy per day were introduced in Model 1. In addition to those factors, mother’s educational level and the total score obtained in the HOME scale at 22 months of age were introduced in Model 2. As can be observed, Model 1 included biomedical factors, while in Model 2 environmental factors were added, in order to test the effect of these types of factors on cognitive development at 22 months of age.

A linear regression analysis was performed using BDI cognitive score at 60 months of age as DV. As independent variables, gender, gestational age (numerical), stay in NICU, Apgar score at 1st minute, mother’s age, and number of cigarettes smoked during pregnancy per day were introduced in Model 1. In addition to those, mother’s educational level and the total score obtained in the HOME scale at 48 months of age were introduced in Model 2. Finally, in Model 3 the BDI cognitive score at 22 months, and the Corsi score at 60 months were added.

## 3. Results

Descriptive results obtained in the two cognitive measures taken at 22 and 60 months of age in the 4 GA groups are shown in Table 2.

In contrast with former studies [70,77], we did not find significant differences between FT and PR children of any GA group in working memory performance. The results of the HOME scale indicate that the children’s families offered similar experiences and opportunities for development to the children, independently of the group.

The results of the 2 (age) × 4 (GA groups) repeated measures ANOVA performed on the results obtained in the BDI at 22 and 60 months of age indicate that there was a highly significant effect of age (intra-subjects differences) on cognitive development (F(1)= 5315.463, *p* < 0.001, *η²* = 0.975). No significant combined effect of age × GA groups was found (F(3)= 0.930, *p* > 0.05, *η²* = 0.020), and no significant difference among GA groups (inter-subjects effects) was found (*F*(3)= 0.923, *p* > 0.05, *η²* = 0.020).

The linear regression analysis performed using BDI cognitive score at 22 months of age (BDI 22) as DV (see Table 3) indicates that the introduction of biomedical variables in Model 1 has a significant effect on cognitive development (*p* < 0.01) and explains 10.2% of the variance (*R**^2^* = 0.102). The variables which have a significant effect (Standardized *β*) are gender and mother’s age at birth. When environmental variables (maternal educational level, and HOME score at 22 months) are added in Model 2, the effect of the independent variables on cognitive development (BDI 22) increments the significance (*p* < 0.001), and the variance explained reaches 0.213 (change in *R*^2^ = 0.111). Now, the variables which have a significant effect are gender and mother’s age at birth (as in Model 1), plus the quality of home environment (HOME total score at 22 months of age).

The linear regression analysis performed using BDI cognitive score at 60 months of age as DV (BDI 60) (see Table 4) indicates that the introduction of the biomedical variables in Model 1 does not have any significant effect on the variance of cognitive development (BDI 60). The variance explained is minimal (*R*^2^ = 0.014), and no single variable reaches significance. When environmental variables (mother’s educational level and HOME score at 48 months) are added in Model 2, the variance explained increases 0.068 (change in *F* is significant), and the model reaches significance (*p* < 0.05). The only variable which reaches a unique significant effect is the HOME score administered at 48 months of age (*p* < 0.05). In Model 3 previous cognitive score (BDI 22) is introduced, together with the Corsi score. Model 3 reaches a highly significant effect on cognitive development measured at 60 months of age (*p* < 0.001), and the introduction of the BDI cognitive score at 22 months of age and the Corsi score significantly incremented the variance explained (Change in *R*^2^ = 0.165). Working memory and previous cognitive score have a unique significant effect (standardized *β*), which was particularly high in the case of BDI 22 previous cognitive score (*p* < 0.001).

## 4. Discussion

The first objective of this research was to check if differences exist in the cognitive development of 4 groups of children with different gestational ages, and, in general terms, healthy. The results of the repeated measures ANOVA performed on the raw scores obtained with the BDI at 22 and 60 months of age clearly indicate that there was no significant difference among the four groups. There was no inter-subjects difference, and no combined effect of age and GA group. The latter result indicates that the trajectories that the children with different GAs follow are similar. As logical, the analysis performed with the raw scores point to a very important age effect (*η*² = 0.975) on cognitive development. The effect of belonging to a different GA group has practically no effect (*η*²= 0.020).

These results reinforce the idea that low risk PT children do not have cognitive delay as compared to FT children [37,40,41], and that GA on its own does not seem to affect cognitive development at least up to the age of 5 years, unless GA is associated with other factors (such as biomedical problems, or unfavorable environmental circumstances) [8,14]. Even comparing the most distant GA groups (FT versus VPT and EPT groups), no significant difference was found. The results found reinforce the idea that healthy condition is an important factor which prevents cognitive delay.

With regard to the second objective, the results of the linear regression analyses showed that the effect of the different predictors changes in relation to the time of measurement of cognitive development. Biological factors seem to have a significant effect on cognitive development measured early in development (22 months of age), particularly gender and mother’s age. The results indicate that boys tend to have lower results than girls [13,49], and that children whose mothers are younger have higher cognitive results during infancy [57]. These effects, however, do not exist when cognitive development is measured later in development, at 60 months of age. This result indicates that the effect of biomedical factors on cognitive achievement descends as children develop [49,59].

In the case of environmental factors, the quality of home environment stands out as the most important factor and it has an effect on cognitive development at 22 months. Therefore, both biological and environmental variables (particularly gender, mother’s age at birth and HOME score) have a significant effect on cognitive development at 22 months of age. In any case the biological and environmental variables introduced in model 2 explain 0.213 of the variance of the BDI cognitive score at 22 months of age, and, therefore, the model reaches significance. The effect of environmental factors is higher in early cognitive development (BDI 22) (Change in *R*^2^ = 0.111) than in later cognitive development (BDI 60) (Change in *R*^2^ = 0.068).

In relation to the explanation of cognitive score at 60 months of age, the biological variables introduced in model 1 have no significant effect and model 1 does not reach significance, while the introduction of environmental variables (particularly the HOME score) in model 2 increment the variance explained to 0.082 and the model reaches significance, reinforcing the predictive role of the HOME scores on cognitive development [27,72].

By far, the variables which have a higher effect on cognitive development at 60 months are the cognitive result obtained earlier in development (BDI 22), and the working memory. The introduction of these independent variables makes the variance explained reach 0.232 (Change in *R*^2^ = 0.165), which is an important effect. The cognitive scores obtained at 22 months of age has the most significant effect, but non-verbal working memory score at 48 months have a moderate effect as well. At the same time these cognitive factors subsume the effect of the quality of home environment, which now, in model 3, does not reach significance.

Therefore, our results coincide with those studies which pointed to the relevance of former cognitive scores in the prediction of later cognitive development [39,73,74,75,76].

A final point, but not of minor relevance, is that gestational age does not have any significant effect on the prediction of cognitive development, indicating that GA is not a predictive factor of cognitive development when serious medical complications are controlled.

## 5. Conclusions

The most important conclusion of this study is that there are no differences in cognitive development among children with different gestational ages if they are healthy. Therefore, low risk PT children do not seem to present cognitive delay in relation to FT children, up to the age of 60 months. Future studies should test if differences might exist later in development.

Biological factors have a modest effect on first cognitive development as measured at 22 months of age, although these factors lose their effect on later cognitive development. Environmental factors (quality of home environment in particular) seem to be more important, and their effect continues over time. The most important predictors of cognitive development at 60 months of age are previous cognitive measures.

## Figures and Tables

**Table 1 ijerph-17-02380-t001:** Descriptive data and comparisons among the 4 GA groups ^a^.

**Chi square**		**≤31** **EPT&VPT**	**32–33** **MPT**	**34–36** **LPT**	**≥37** **FT**	**χ^2^**	**df**
Gender (frequency)	Female	19	14	32	21	2.64	3
	Male	24	22	26	23		
Maternal educational level (frequency)	Basic	8	6	7	12	12.44	6
	High school	16	16	37	15		
	University	19	14	14	16		
Stay in NICU (frequency)	No stay	1	4	31	40	120.84 *****	6
	1–15 days	13	22	23	2		
	>15 days	29	10	4	1		
**ANOVA**		**≤31** **EPT&VPT**	**32–33** **MPT**	**34–36** **LPT**	**≥37** **FT**	**F**	**df**
Mother’s age	Mean	34.88	34.72	33.24	31.98	4.23 ****	179
	SD	5.06	4.24	3.63	4.41		
Cigarettes	Mean	0.77	0.67	1.16	1.05	0.249	179
	SD	2.53	2.39	3.61	3.34		
Apgar score 1st minute	Mean	7.14	8.33	8.31	8.14	9.73 *****	179
	SD	1.58	0.95	0.99	1.20		
HOME score at 22 months	Mean	37.30	39.33	38.24	38.70	1.64	179
	SD	4.54	3.85	4.35	3.97		
HOME score at 48 months	Mean	48.14	49.12	49.40	49.97	1.86	144
	SD	3.35	3.87	3.48	2.46		
Corsi score at 60 months	Mean	10.50	9.25	11.78	10.64	0.540	127
	SD	7.42	8.30	8.47	7.60		

^a^ Chi square in the upper part, and one-way ANOVA in the lower part. ** *p* value < 0.01, *** *p* value < 0.001.

**Table 2 ijerph-17-02380-t002:** Descriptive results of cognitive development by gestational age groups.

	GA in Weeks	n	Mean	SD	Range
BDI cognitive score at 22 months of age	≤31	44	26.55	2.98	20–34
	32–33	36	26.83	3.22	22–33
	34–36	58	26.71	3.71	12–36
	≥37	43	27.58	4.02	22–39
	Total	181	26.90	3.53	12–39
BDI cognitive score at 60 months of age	≤31	33	84.48	8.54	51–93
	32–33	31	85.58	5.34	73–95
	34–36	42	82.40	15.64	11–94
	≥37	33	85.91	3.18	78–92
	Total	139	84.44	10.01	11–95

**Table 3 ijerph-17-02380-t003:** Linear Regression analysis: predictors of BDI cognitive scores at 22 months.

Predictors (IV)	Standardized *β*	*R^2^*	Change in *R^2^*	Change in *F*	*F*(df)
Model 1		0.102	0.102	3.147 **	3.147 (6,166)
Gender	0.225 **				
GA in weeks	0.129				
Stay in NICU	−0.038				
Apgar score 1st minute	0.085				
Mother’s age	0.197*				
Cigarettes per day	−0.140				
Model 2		0.213	0.111	11.536 ***	5.543 (2,164)
Gender	0.188 **				
GA in weeks	0.144				
Stay in NICU	0.001				
Apgar score 1st minute	0.061				
Mother’s age	0.171 *				
Cigarettes per day	0.044				
Mother’s educational level	0.045				
HOME score at 22 months	0.341 ***				

* *p* value < 0.05, ** p value < 0.01, *** *p* value < 0.001.

**Table 4 ijerph-17-02380-t004:** Linear Regression: predictors of BDI cognitive scores at 60 months.

Predictors (IV)	*Standardized β*	*R^2^*	Change in *R^2^*	Change in *F*	*F*(df)
Model 1		0.014	0.014	0.283	0.283 (6,119)
Gender	−0.042				
GA in weeks	−0.029				
Stay in NICU	−0.082				
Apgar score 1st minute	0.064				
Mother’s age	−0.027				
Cigarettes per day	0.009				
Model 2		0.082	0.068	4.311 *	1.302 (8,117)
Gender	−0.036				
GA in weeks	−0.010				
Stay in NICU	−0.028				
Apgar score 1st minute	0.061				
Mother’s age	0.014				
Cigarettes per day	0.002				
Mother’s educational level	0.073				
HOME score at 48 months	0.241 *				
Model 3		0.232	0.165	11.259 ***	3.476 (10,115)
Gender	0.038				
GA in weeks	−0.045				
Stay in NICU	−0.034				
Apgar score 1st minute	0.037				
Mother’s age	−0.077				
Cigarettes per day	0.040				
Mother’s educational level	0.116				
HOME score at 48 months	0.103				
BDI score at 22 months	0.334 ***				
Corsi score at 60 months	0.200*				

* *p* value < 0.05, ** *p* value < 0.01, *** *p* value < 0.001.

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
