# Peer review of "A Follow-Up Study of Cognitive Development in Low Risk Preterm Children"

_ijerph, 2020, doi:10.3390/ijerph17072380_

Round 1
Reviewer 1 Report
The authors present a longitudinal study on the cognitive development at 22 and 60 months of age for 1 group of full-term and 3 groups of low risk preterm children with different gestational ages (GA). Based on their results they conclude that GA does not affect cognitive development when preterm children are healthy.
I have only minor suggestions:
At the beginning of the Introduction please provide a definition for “full-term” (e.g. GA of 37 weeks or above).
Please also define “low risk preterm” in line 49.
Please also explain the term “Apgar scores” in line 65.
There is a typo in line 72: Change “neuropshychological” into “neuropsychological”.
There is a typo in line 99: Change “PK” into “PT” children.
In the Procedures part in line 163, the authors write, “The children were assessed when they were 22, 48 and 60 months of age.” Did they also assess at 48 months of age? If so, what were the reasons why they don’t report the data? The next sentence states that “Corrected age for PT children was used at 22 and 30 months of age”. What does that mean? Why at 30 months of age?
Line 173: HOME is a questionnaire? Does the examiner ask the parents? Please shortly describe the procedure.
Line 218: 48 months of age are not shown in Table 2.
Line 234: Please explain the effect of “gender”.
Line 265: Please restrict this conclusion “does not seen to affect cognitive development…” at least up to the age of 5 years (60 months), as this was the period of your examination.
Author Response
Definition of Full-term children is provided in line 134 (Participants section). If we introduce this information in the introduction, the number of words would exceed the maximum allowed for the introduction.
Definition of low risk preterm children is already provided in the text (see line 54), when we say "without major health problems". We give a wider description of the low risk condition in the description of the participants.
The term Apgar score is now explained (see lines 70-73): "Apgar test is administered to newborn children in the first minute of life (and also at minute 5 and 10, if necessary) and provides a measure of the health state of the children in 5 vital areas, offering a score from 0 to 10."
Neuropshychological has been changed into neuropsychological (line 79 of the revision).
PR has been corrected for PT (line 109).
The children were assessed at 22, 48 and 60 months of age. At 48 months of age the children were assessed with the HOME scale, and the results are reported in Table 1.
There was a mistake on the next paragraph: "Corrected age for PT children was used at 22 and 30 months of age". Now it appears as "Corrected age for PT children was used at 22 months of age" (line 74).
The procedure to gather information through the HOME scale is now shortly described in lines 185-187: “The information is gathered through personal interview to mothers and through observation of the children’s homes and the interactions between the children and the mothers during the interview session.”
There was a mistake in lines 218-219. Now it appears as "Descriptive results obtained in the two cognitive measures taken at 22 and 60 months of age in the 4 GA groups are shown in Table 2." (Lines 230-231 of the revised version).
The effect of gender, as well as the effect of mother's age, is explained in the Discussion: "The results indicate that boys tend to have lower results than girls [13,49], and that children whose mothers are younger have higher cognitive results during infancy [57]." (Lines 286-288 of the revised version).
As suggested by the reviewer we included the text "at least up to the age of 5 years" (line 278 of the revised version) to limit the extent of the claim.
Reviewer 2 Report
A follow-up study of cognitive development in low risk preterm children
Perez-Pereira, Fernandez, Gomez-Taibo, Maritnez-Lopez, and Arce
This paper presents a longitudinal study comparing cognitive development in full term versus healthy preterm children of different gestational ages. Cognitive development was measured using the BDI and predictive factors were assessed at 22 and at 60 months of age, accounting for biomedical, environmental and individual factors. Overall the authors found that cognitive development at 22 months was the best predictor of cognitive development at 60 months; gestational age did not predict cognitive development in this case, where all infants were healthy despite differences in how early they were born.
Major Comments:
While the paper tests an interesting idea the rationale needs to be clearer and better motivated and the style in which the paper is written needs extensive revision. Some paragraphs in the paper are only one or two sentences and need to be combined and flow more cohesively. The theme of each paragraph needs to be clearer and more directly be tied back to the main point of the study.
For example:
(1) The authors mention that parental education level is an important predictor of cognitive development. They need to discuss how parental education was determined and how their own measures relate to already published literature, Do the authors use the same measures? Is it only maternal and not paternal education that matters? How difficult is it to disentangle parental education level from SES?
(2) The authors mention issues with the use of APGAR scores, but need to better motivate the cognitive measures they chose to use instead.
(3) Many of the previous studies the authors mention look at the influence of gestational age on cognitive development, but later development, such as in children 2-7 years of age, 4 and 15 years of age, or look at the influence of other factors on cognitive development, such as smoking or parental education, also later in development, such as in children, 4 or 5 or even 18 years of age. They need to more clearly elaborate on known evidence int he specific age range they are interested in and, more importantly, consider if differences might not exist at 60 months but might arise later in development.
Minor Comments:
In Table 1, it would help the reader if labels included not just gestational age but also the categories defined by the authors (VPT, EPT, MPT, LPT, FT).
Table 1 should also include the measure BW.
Author Response
The paper had already been revised by a native expert speaker of English. However, the entire manuscript has been revised again a few changes were introduced in order to improve the linking of sentences. In any case, we ask for comprehension the reviewer. We are not native speakers of English, and our style cannot be as perfect as the style of a native speaker. In addition, since the measures taken to fight against the Covid-19, the situation in our country does not permit social contact and we could not meet the native expert again.
1) In lines 148-149 (revised version; and lines 138-139 of the original paper) we describe how mother`s education was computed. In addition, we did the same for stay in NICU (line 150). Obviously, educational systems around the world change, and the scales used to measure educational level also change. In Spain we do not have standard criteria to compute maternal education, as, for example, exists in the USA.
In our study we only took into consideration maternal education (and not father’s education). First, because it seems to be the most significant measure, and secondly, because we tried to reduce the number of variables introduced in the analysis.
Certainly, maternal education and SES are closely linked. The purpose of this study is not to investigate the relationships between SES and maternal education. We took into consideration maternal education. There is great variation in the consideration of these measures (mother’s education, father’s education, SES) across studies.
Descriptive information on the maternal educational level and the length of stay in NICU of the participants are also included in Table 1.
2) In relation to the Apgar score, a new paragraph has been introduced (see lines 70-73), as requested by reviewer 1. In any case, to say the truth, I cannot understand the point reviewer 2 rises (“The authors mention issues with the use of APGAR scores, but need to better motivate the cognitive measures they chose to use instead”).
3) The revision of the studies on the cognitive development of preterm children and the possible predictive factors was carried out with special attention to the range of the ages studied in this investigation, although studies carried out with children of older ages were also introduced. In every case, the specific age of the participants is informed. We mention also studies carried out with children older than 60 months because it is relevant to check if the effects of certain variables on the cognitive development of these children are also observed in the age range we studied.
Now it is remarked that the absence of differences between healthy PT children and FT children in cognitive development is restricted to the age of 5 years (lines 278, and 320-321 of the revised version), and that differences might exist later in development (line 321).
Minor revisions:
1) As suggested by the reviewer, labels (VPT, EPT, MPT, LPT, FT) were introduced in Table 1.
2) In table 1 BW was not introduced since this variable is not analyzed. Extremely high correlations between GA and BW were found in our sample. For this reason, we decided to use only one of them (GA).